# Association between Adverse Childhood Experiences and Time Spent Playing Video Games in Adolescents: Results from A-CHILD Study

**DOI:** 10.3390/ijerph181910377

**Published:** 2021-10-02

**Authors:** Satomi Doi, Aya Isumi, Takeo Fujiwara

**Affiliations:** 1Department of Global Health Promotion, Tokyo Medical and Dental University (TMDU), Bunkyo 113-8519, Japan; doi.hlth@tmd.ac.jp (S.D.); isumi.hlth@tmd.ac.jp (A.I.); 2Japan Society for the Promotion of Science, 5-3-1 Kojimachi, Tokyo 102-0083, Japan

**Keywords:** adolescent, adverse childhood experience, gaming, peer isolation

## Abstract

Background: Excessive time spent playing video games is associated with adverse health outcomes in adolescents. Although poor child–parent relationship and social relations with peers are considered as possible predictors, little is known as to whether adverse childhood experiences (ACEs) are associated with time spent playing video games. The aim is to examine the association between ACEs and time spent playing video games in adolescents. Methods: We used pooled data from the Adachi Child Health Impact of Living Difficulty (A-CHILD) study in 2016 and 2018, which is a population-based cross-sectional study in Adachi City, Tokyo, Japan (*N* = 6799, 4th, 6th, and 8th-grade students). Adolescents answered questionnaires examining the time spent playing video games, per day, on weekdays (“less than 1 h”, “less than 3 h”, and “more than 3 h”) and ACEs (eight types). Results: The results of the ordinal logistic regression analysis showed a positive association between ACE total score and time spent playing video games after adjusting for covariates (1 ACE: OR = 1.28, 95% CI = 1.10–1.48; 2 ACEs: OR = 1.25, 95% CI = 1.06–1.48; 3 + ACEs: OR = 1.44, 95% CI = 1.14–1.82, *p* for trend < 0.001). Regarding each type of ACE, the experiences of single parenthood, parental history of psychiatric disorders, and peer isolation were independently positively associated with time spent playing video games. Conclusions: Health policy to address ACEs might be important to shorten the time spent playing video games.

## 1. Introduction

Excessive time spent playing video games in children and adolescents has been a public health issue globally. The American Psychiatric Association has defined “Internet gaming disorder (IGD)” which refers to “persistent and recurrent use of the Internet to engage in games, often with other players, leading to clinically significant impairment or distress” in the Diagnostic and Statistical Manual, Fifth Edition (DSM-5) [1]. The World Health Organization (WHO) also included Gaming Disorder (GD) into the 11th Revision of the International Classification of Diseases (ICD-11) [2]. A long duration of video game play is a risk factor for IGD [3,4]. Furthermore, excessive time spent playing video games is associated with adverse physical and psychological health outcomes in adolescents such as overweight and obesity [5,6], poor sleep quality [6,7], depression and anxiety [8,9], increased aggression [10], and reduced prosocial behaviors [10]. To prevent these negative physical and psychological health outcomes including IGD, identifying predictors for excessive time spent playing video games is needed.

Previous studies have shown several predictors for excessive time spent playing video games in adolescents including problematic video gaming and IGD. The systematic review, which identified predictors for problematic video gaming in adolescents, showed that a poor quality of parent–child relationship (i.e., attachment) might predict problematic video gaming [11]. Another review paper investigating predictors for IGD in children and adolescents also that showed poor parent–child relationships, family dysfunction such as paternal divorce and single parenthood, and childhood maltreatment might predict IGD [11,12]. In addition to these family factors, poor social relations with peers such as peer isolation or peer victimization is associated with problematic video gaming [13,14,15]. However, as these factors can be comorbid, the cumulative effect of these childhood adverse experiences (ACEs) needs to be assessed, along with the independent association to longer time periods spent playing video games.

ACE is defined as childhood experiences before the age of 18 years including parental loss, household dysfunction such as parental divorce, and child maltreatment such as physical abuse and neglect [16]. A recent study has added peer isolation and low household income to the original ACE scale [17]. In the literature related to addictive behaviors such as substance use, alcohol use, tobacco use, and gambling, studies have shown that an increased number of ACEs is associated with addictive behaviors in adolescents [18,19,20]. A previous study of gamers reported a positive association between ACEs and problematic video gaming [21]. However, little is known about the association between ACE and time spent playing video games among population-based adolescence, the sensitive period for addictive behavior [22]. This study aims to examine the association in adolescents between ACE and amount of time spent playing video games.

## 2. Materials and Methods

### 2.1. Participants

We used pooled data of the Adachi Child Health Impact of Living Difficulty (A-CHILD) study in 2016 and 2018, which is a population-based cross-sectional study in Adachi City, Tokyo, Japan. The details of the A-CHILD study protocol are available from the protocol paper of the A-CHILD [23]. In 2016, anonymized self-reported questionnaires with unique IDs were distributed to 1994 adolescents (in fourth, sixth, and eighth grades) in nine elementary schools and seven junior high schools. Schools were selected based on geographical and socioeconomic representation. In 2018, the questionnaires were distributed to 6625 adolescents in all 69 elementary school fourth grades, nine representative elementary schools sixth grades, and seven representative junior high school second grades. Both adolescents and caregivers completed the questionnaires. For questionnaires to be deemed valid, they needed: (1) informed consent; (2) at least one question to have a response; and (3) the respondent was to have completed another survey, the Study Attitude survey, which was conducted by Adachi City Board of Education. A total of 1652 valid questionnaires was received (response rate = 82.0%) in 2016 and 5382 (response rate = 81.5%) in 2018. Among the valid responses, the participants who missed exposure and outcome variables in this study (i.e., ACEs and time spent playing video games) were excluded. The analytical sample included 6799 adolescents (fourth grade: *N* = 4654; sixth grade: *N* = 1016; eighth grade: *N* = 1129) (Figure 1).

### 2.2. Measurements

ACEs consist of the following eight types, which were based on a previous study [17]: single parent, physical and psychological abuse, neglect, parental psychiatric history, witness to domestic violence between parents, low household income, and peer isolation. Instead of using the ACE scale, we assessed ACEs using the questions related to each type of ACE from both the caregiver and adolescent questionnaires. In this study, seven types, except peer isolation (i.e., single parent, physical and psychological abuse, neglect, parental psychiatric history, witness to domestic violence between parents, and low household income), were assessed on the caregiver’s questionnaire in order to avoid the adolescent’s psychological invasiveness. For each type of ACE, we created a binary variable: a score of “0” denoted that adolescents have never experienced (i.e., “no” response), and a score of “1” denoted that adolescents have ever experienced (i.e., “yes” response).

Single parenthood was assessed on the caregiver’s questionnaire with a question about living with family members. When family members living together did not included a mother or father, single parenthood was coded as “1”. Parental history of psychiatric disorder was assessed by two questions (i.e., maternal history of psychiatric disorders and paternal history of psychiatric disorders) on the caregiver’s questionnaire. When the caregiver reported either maternal or paternal history of psychiatric disorders, parental history of psychiatric disorder was coded as “1”. Physical and psychological abuse, witness to domestic violence, and neglect were assessed via the caregiver questionnaires using seven items on a scale of 1 = “often”, 2 = “sometimes”, 3 = “rarely”, and 4 = “not at all.” Physical abuse was assessed from two questions: “hit the child’s body (buttocks, hand, head, or face)”, in which the responses were dichotomized with “often” equated to a “yes” response, and “beat the child,” in which the responses were dichotomized with “rarely”, “sometimes”, or “often”, which equated to a “yes” response. When either item was classified as “yes,” physical abuse was coded as “1”. Psychological abuse was assessed by two questions: “yell at the child,” in which the responses were dichotomized with “often” equated to a “yes” response, and “insult the child repeatedly”, in which the responses were dichotomized with “sometimes” or “often”, which equated to a “yes” response. When either item was classified as “yes,” psychological abuse was coded as “1”. Witness to domestic violence was assessed by the question “have a big fight in front of the child,” in which the responses were dichotomized with “sometimes” or “often”, which were both equated to a “yes” response. Neglect was assessed by two questions: “shut the child outside” and “do not feed the child,” in which the responses were dichotomized with “rarely”, “sometimes”, or “often”, which equated to a “yes” response. When either item was classified as “yes”, neglect was coded as “1”. This coding was developed in a previous study [24]. Low household income was assessed in the caregiver questionnaire, in which the response “3 million yen or less” was coded as “1”. Peer isolation was assessed in the adolescent questionnaire with the question “how many friends can you talk to about your worries/troubles?” The responses were “0”, which was coded as “1.”

The amount of time spent playing video games on weekdays was measured by the adolescent questionnaires. Adolescents assessed the item on a scale of 1 = “not at all”, 2 = “30 min”, 3 = “1 h”, 4 = “1 h and a half”, 5 = “2 h”, 6 = “2 h and a half”, 7 = “3 h”, and 8 = “more than 3 h”. In this study, “not at all”, “30 min”, and “1 h” were categorized as “1” (less than 1 h), “1 h and a half”, “2 h”, “2 hours and a half”, and “3 h” were categorized as “2” (less than 3 h), and “more than 3 h” was categorized as “3” (more than 3 h).

Covariates were the child’s sex, number of siblings, grade in school, and maternal age, which were measured via the questionnaire. These variables were selected as covariates because the child’s sex and maternal age were used as confounders in previous studies that examined the association between ACEs and addictive mobile phone use [25] and the association between family function and problematic Internet use [26]. Furthermore, the number of siblings is associated with ACEs [27] and gaming [28,29].

### 2.3. Ethics

This study was conducted according to the guidelines of the Declaration of Helsinki and approved by the institutional review board of the Tokyo Medical and Dental University (M2016-284).

### 2.4. Statistical Analysis

In the analyses, the cumulative ACE total score was calculated using all eight types and collapsed as 0, 1, 2, and 3+ due to the small sample size for 4+ cases (N = 149, 2.2%). Each type of ACE was used in the separate analysis. Ordinal logistic regression analysis was performed to examine the association of the cumulative number of ACEs and each type of ACE with time spent playing games (i.e., “1 h or less”, “3 h or less”, and “more than 3 h”). After estimating the crude model, we examined the association between the cumulative number of ACEs and time spent playing games, adjusting for the child’s sex, grade in school, number of siblings, maternal age, and child’s school (Model 1). As for each type of ACE, Model 2 adjusted for child’s sex, grade, number of siblings, maternal age, and child’s schools. Model 3 included types of ACEs that were significant in Model 2. All analyses included the missing data as dummy variables and were weighted for the number of responses in each grade. We conducted all analyses using STATA version 15.0 SE. These analyses were not pre-registered.

## 3. Results

### 3.1. The Distribution of Characteristics

Table 1 shows the distribution of characteristics by grade among the participants. About half of the adolescents, in all grades, had one or more siblings. Approximately 90% of mothers involved in the research were 35 years old or older. A total of 60.2% of adolescents had no ACEs, 19.4% had one ACE, 13.9% had two ACEs, and 6.5% had three or more ACEs. In terms of time spent playing video games, per day, on weekdays, 62.9% of adolescents played video games 1 h or less per day, 27.0% played video games 3 h or less per day, and 10.1% played video games more than 3 h per day. As the adolescent participant’s age increased, those who played video games more than 3 h per day also increased (4th grade = 8.2%, 6th = 10.1%, 8th = 17.6%, *p* < 0.001).

Table 2 shows the distribution of each type of ACE by grade in school. Among all of the participants, approximately 15% of adolescents had experiences of single parenthood, 13% had the experience of peer isolation, and 11% had experiences of low household income. We also show the relationship between each type of ACE in Table 3 in which high comorbidity was found. For example, 61.5% of adolescents who had single parents also experienced low household income; and 41.6% of adolescents who experienced psychological abuse from parents also experienced physical abuse from their parents.

### 3.2. The Association between ACEs and Time Spent Playing Video Games

The results of the ordinal logistic regression analysis are shown in Table 4. The cumulative number of ACEs was found to have a positive association with time spent playing video games, after adjusting for covariates (*p* for trend < 0.001). After adjusting for covariates, the odds ratio (OR) of ACEs was significant (1 ACE: OR = 1.28, 95% CI = 1.10–1.48; 2 ACEs: OR = 1.25, 95% CI = 1.06–1.48; 3+ ACEs: OR = 1.44, 95% CI = 1.14–1.82, *p* for trend < 0.001). Regarding each type of ACE, the ORs of single parenthood (OR = 1.60, 95% CI = 1.35–1.90), parental history of psychiatric disorders (OR = 1.68, 95% CI = 1.35–2.10), peer isolation (OR = 1.31, 95% CI = 1.10–1.54), and lower household income (OR = 1.51, 95% CI = 1.24–1.83) were significant after adjusting for covariates. In Model 3, which added not only covariates, but also types of ACE that were associated with time spent playing video games (i.e., single parenthood, parental history of psychiatric disorders, peer isolation, and lower household income), it was found that single parenthood (OR = 1.55, 95% CI = 1.21–1.97), parental history of psychiatric disorders (OR = 1.62, 95% CI = 1.28–2.04), and peer isolation (OR = 1.32, 95% CI = 1.10–1.58) were independently associated with time spent playing video games.

## 4. Discussion

The current study found a positive relationship between the cumulative number of ACEs, which included the experiences of peer isolation and lower household income, and more time spent playing video games among Japanese adolescents. Specifically, adolescents living in single-parent families, adolescents who had parents with a history of psychiatric disorders, and adolescents with the experiences of peer isolation were more likely to spend time playing videogames per day on weekdays than those without such experiences. These three types of ACE were independently associated with longer amounts of time spent playing video games. Even though there is no consistent and standardized definition of excessive time spent playing video games, a previous study showed that adolescents who categorized as clinical group of IGD spent 27.8 h (SD = 13.3), those at risk of IGD spent 20.8 h (SD = 16.3), and those without problems spent 4.8 h (SD = 8.1) playing video games per week [30]. The categorical variable of time spent playing video games used in this study (i.e., “1 h or less”, “3 h or less”, and “more than 3 h”) is broadly similar to the average of time spent playing video games among IGD classification groups.

Our findings, which had a positive relationship between ACEs and time spent playing video games, were consistent with the results of a previous study focusing on problematic video gaming among gamers [21]. ACEs might already impact the amount of time adolescents spend playing video games, which would be consistent with findings from previous studies examining the adverse impact of ACEs and other addictive behaviors in adolescence [18,19,20]. Furthermore, the results for each type of ACE, single parenthood, paternal history of psychiatric disorders, and peer isolation were associated with longer time spent playing video games, which is consistent with previous longitudinal and cross-sectional studies [31,32,33]. However, child maltreatment (physical abuse, psychological abuse, witness to domestic violence between parents, and neglect in this study) showed no significant association with time spent playing video games. Although a few studies have examined the association between child maltreatment and problematic gaming among children and adolescents [31,34,35], two studies focusing on physical abuse showed no association [31,34]. In this study, we might assess slight-to-moderate child maltreatment rather than severe child maltreatment. Moreover, we could not assess sexual abuse. Thus, the caregivers who reported child maltreatment as a form of strict discipline might also be strict in dictating the amount of time the child is allowed to play video games. Further studies need to examine the association between severe child maltreatment and time spent playing video games.

The positive association between the cumulative number of ACEs and time spent playing video games in adolescents may be explained by the following psychological and biological mechanisms. In terms of psychological mechanism, attachment style or child–parent relationship may mediate the association between ACEs and longer time spent playing video games. The previous systematic reviews regarding adolescent problematic gaming indicate that a poor parent–child relationship is a critical risk factor for problematic gaming including IGD [11,36,37]. A poor parent–child relationship may lead to less time to communicate with family members, which may increase the amount of time spent playing video games [38]. Additionally, a poor parent–child relationship may affect parental supervision and care for the child. Thus, adolescents who live in single-parent families or have parents with a history of psychiatric disorders might be less likely to be supervised and cared for by their parents [31]. Moreover, Zhu et al. [39] found that poor parent–child relationships had an indirect effect on IGD via poor school connectedness. Additionally, another study indicated that children with lower levels of social integration into school classes were more likely to be at a higher risk of video game addiction [31], as peer isolation was associated with longer time spent playing video games in this study. Children with poor peer relationships might be more likely to engage in virtual worlds instead of real life.

In terms of biological mechanism, structural and functional brain alterations [40] might mediate the association between ACE and time spent playing video games. For example, Herzog and Schmahl [40] indicated that ACEs led to structural and functional alterations of the anterior cingulate cortex (ACC) and hippocampus, which are related to the ability of error monitoring [41] and self-regulation [42]. Adolescents with ACEs showed lower levels of self-regulation skills, which were assessed using a flanker task [43]. Numerous studies have found dysfunction of self-regulation skills to be associated with IGD among adolescents [44,45]. Furthermore, as a bio-psychological mechanism, ACEs may induce biological changes such as stress-related inflammation including C-reactive protein (CRP), interleukin 6 (IL-6), and soluble urokinase plasminogen activator receptor (suPAR) [46], structural and functional brain alterations including ACC, hippocampus, and amygdala [40], which may lead to mental health problems. As the interaction with poor mental health status and longer time spent playing games has been found [36], we cannot assure a causal relationship. Instead, we can merely recognize that adolescents with mental health problems such as depression due to ACEs may be more likely to spend time playing games.

The current study has the following limitations. First, we cannot suggest a causal relationship between ACEs and time spent playing games due to the cross-sectional study, that is, reverse causation is likely. As for peer isolation, adolescents who spend longer amounts of time playing games might struggle with having friends with whom they can consult. Further longitudinal studies that examine the causal relationships are needed. Second, sexual abuse was not included in ACEs. In Japan, the prevalence of sexual abuse is low (0.5%) [47] and sexual abuse is considered to be a sensitive issue [48]. Thus, asking about sexual abuse may cause a lower response rate [49]. Third, almost all types of ACEs were assessed on the caregiver’s questionnaire because we avoided the adolescent’s psychological invasiveness, which might cause information bias. There might be discrepancies between the caregiver’s and child’s reports related on ACEs such as abuse and neglect. Fourth, there might be sampling bias related to responses to the questionnaire due to a lower response rate from adolescents with ACEs and their caregivers. Fifth, the study did not assess the time spent playing games on weekends due to the logistics of implementing the study. Although previous studies have confirmed the correlation between weekday and weekend time spent playing games, further study should include the time spent playing games on the weekend.

To prevent longer times spent playing video games in adolescents and eventually to prevent IGD and GD in children and adolescents, the development of self-regulation skills might need to be addressed in schools [50], targeting those who experienced ACEs, especially children and adolescents with single parents, parental history of psychiatric disorders, and peer isolation. Additionally, providing opportunities to increase time spent on other leisure-time activities, aside from gaming for adolescents with ACEs, may be helpful [51,52,53]. For parents who are single or have a history of psychiatric disorders, education on how to supervise their child’s use of games might be needed [54].

## 5. Conclusions

We found that the cumulative number of ACEs was associated with longer time spent playing video games in adolescents. Regarding the types of ACEs, single parenthood, parental history of psychiatric disorders, and peer isolation were independently associated with longer times spent playing video games. To develop a population approach program designed to prevent longer times spent playing video games in adolescents, it may be needed to assess not only family background and relationships with their peers.

## Figures and Tables

**Figure 1 ijerph-18-10377-f001:**
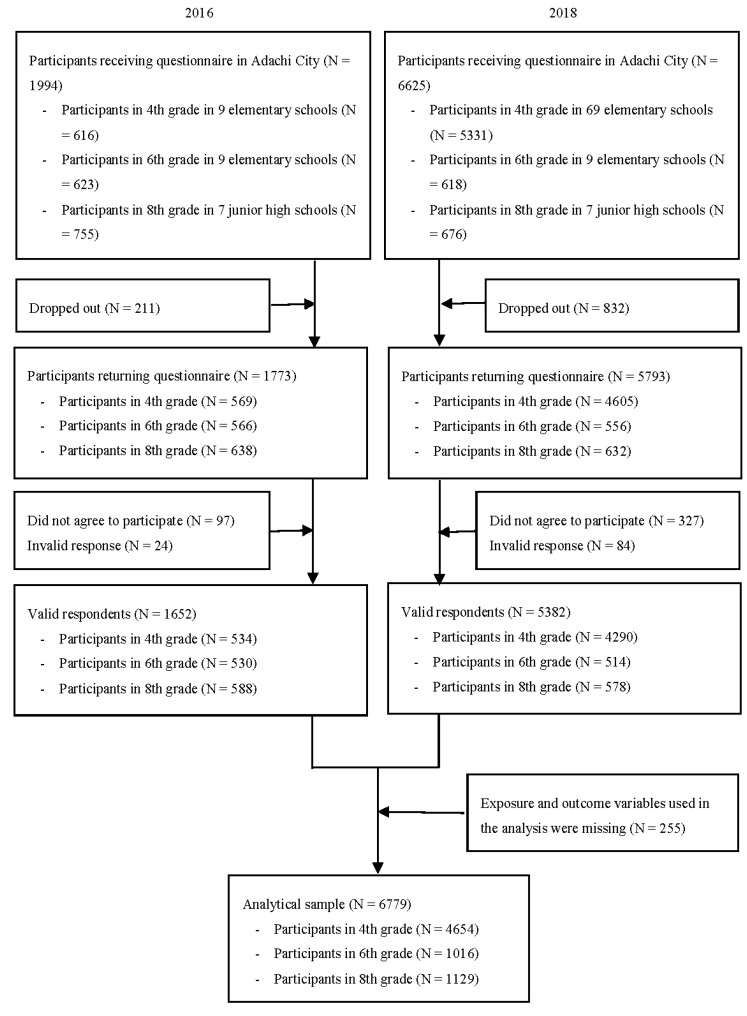
Participants flow chart.

**Table 1 ijerph-18-10377-t001:** Characteristics of the sample (*n* = 6799).

		Total		Grade					
				4th (*n* = 4654; 68.5%)		6th (*n* = 1016; 14.9%)		8th (*n* = 1129; 16.6%)	
		*n* or Mean	% or SD	*n* or Mean	% or SD	*n* or Mean	% or SD	*n* or Mean	% or SD
Child’s sex	Male	3375	49.6	2342	50.3	491	48.3	542	48.0
	Female	3422	50.3	2311	49.7	525	51.7	586	51.9
	Missing	2	0.1	1	0	0	0	1	0.1
Number of siblings	No sibling	3334	49.0	2283	49.1	483	47.5	568	50.3
	1	1725	25.4	1146	24.6	274	27.0	305	27.0
	2	351	5.2	257	5.5	39	3.8	55	4.9
	3+	92	1.4	61	1.3	15	1.5	16	1.4
	Missing	1297	19.1	907	19.5	205	20.2	185	16.4
Maternal age	<30	48	0.7	44	0.9	4	0.4	0	0
	30–34	563	8.3	455	9.8	64	6.3	44	3.9
	35–39	1508	22.2	1162	25.0	208	20.5	138	12.2
	40–44	2460	36.2	1687	36.2	361	35.5	412	36.5
	45+	1967	28.9	1145	24.6	341	33.6	481	42.6
	Missing	253	3.7	161	3.5	38	3.7	54	4.8
Cumulative ACE total score	0	4093	60.2	2719	58.4	659	64.9	715	63.3
	1	1318	19.4	894	19.2	190	18.7	234	20.7
	2	946	13.9	688	14.8	120	11.8	138	12.2
	3+	442	6.5	353	7.6	47	4.6	42	3.7
Time spent playing video game per day (weekdays)	Not at all	1553	22.8	1126	24.2	212	20.9	215	19.0
	30 min	1494	22.0	1124	24.2	198	19.5	172	15.2
	1 h	1228	18.1	889	19.1	169	16.6	170	15.1
	1 h and a half	714	10.5	466	10.0	132	13.0	116	10.3
	2 h	569	8.4	338	7.3	104	10.2	127	11.2
	2 h and a half	346	5.1	212	4.6	66	6.5	68	6.0
	3 h	209	3.1	115	2.5	32	3.1	62	5.5
	More than 3 h	686	10.1	384	8.3	103	10.1	199	17.6

**Table 2 ijerph-18-10377-t002:** Distribution of categories of adverse childhood experiences (ACEs).

	Total		4th		6th		8th	
	*n*	%	*n*	%	*n*	%	*n*	%
1. Single parenthood	985	14.5	599	12.9	157	15.5	229	20.3
2. Parental history of psychiatric disorders	505	7.4	354	7.6	59	5.8	92	8.1
3. Physical abuse from parents (hit, slap)	616	9.1	461	9.9	76	7.5	79	7.0
4. Psychological abuse from parents (verb)	294	4.3	213	4.6	41	4.0	40	3.5
5. Witness of domestic violence between parents	291	4.3	201	4.3	47	4.6	43	3.8
6. Neglect from parents (out, unfed)	543	8.0	423	9.1	54	5.3	66	5.8
7. Peer isolation	875	12.9	679	14.6	99	9.7	97	8.6
8. Lower household income (<3,000,000)	750	11.0	486	10.4	119	11.7	145	12.8

**Table 3 ijerph-18-10377-t003:** Relationships between categories of adverse childhood experiences (ACEs).

	N and Percent (%) Exposed to Another ACEs															
	1. Single Parenthood		2. Parental History of Psychiatric Disorders		3. Physical Abuse from Parents (Hit, Slap)		4. Psychological Abuse from Parents (Verb)		5. Witness of Domestic Violence between Parents		6. Neglect from Parents (Out, Unfed)		7. Peer Isolation		8. Lower Household Income (<3,000,000)	
	*n*	%	*n*	%	*n*	%	*n*	%	*n*	%	*n*	%	*n*	%	*n*	%
1. Single parenthood			112	11.4	119	12.4	46	4.8	30	3.1	102	10.6	140	14.7	487	61.5
2. Parental history of psychiatric disorders	112	22.2			67	13.4	48	9.6	39	7.8	60	12.0	70	14.1	99	22.1
3. Physical abuse from parents (hit, slap)	119	19.3	67	10.9			122	19.9	71	11.6	174	28.3	93	15.6	105	20.0
4. Psychological abuse from parents (verb)	46	15.7	48	16.3	122	41.6			67	22.9	82	27.9	58	20.2	43	16.5
5. Witness of domestic violence between parents	30	10.3	39	13.4	71	24.4	67	23.0			60	20.6	46	16.1	43	17.1
6. Neglect from parents (out, unfed)	102	18.8	60	11.1	174	32.2	82	15.2	60	8.1			90	17.0	88	18.7
7. Peer isolation	24	26.4	70	8.0	93	10.8	58	6.7	46	5.3	90	10.5			119	15.5
8. Lower household income (<3,000,000)	487	64.9	99	13.2	105	14.4	43	5.9	43	5.9	88	12.0	119	16.3		

**Table 4 ijerph-18-10377-t004:** The associations between ACEs and time spent playing video games.

		Crude	Model 1	Model 2	Model 3
		OR (95% CI)	OR (95% CI)	OR (95% CI)	OR (95% CI)
ACE total score	0	Ref	Ref		
	1	1.29 (1.12–1.48)	1.28 (1.10–1.48)		
	2	1.49 (1.28–1.75)	1.25 (1.06–1.48)		
	3+	1.67 (1.34–2.06)	1.44 (1.14–1.82)		
		*p* for trend < 0.001	*p* for trend < 0.001		
Single parenthood	No	Ref		Ref	Ref
	Yes	1.58 (1.35–1.84)		1.60 (1.35–1.90)	1.55 (1.21–1.97)
Parental history of psychiatric disorders	No	Ref		Ref	Ref
	Yes	1.64 (1.34–2.02)		1.68 (1.35–2.10)	1.62 (1.28–2.04)
Physical abuse from parents (hit, slap)	No	Ref		Ref	
	Yes	1.05 (0.86–1.27)		0.93 (0.76–1.14)	
Psychological abuse from parents (verb)	No	Ref		Ref	
	Yes	1.04 (0.79–1.37)		1.05 (0.78–1.41)	
Witness of domestic violence between parents	No	Ref		Ref	
	Yes	1.08 (0.83–1.39)		1.14 (0.87–1.50)	
Neglect from parents (out, unfed)	No	Ref		Ref	
	Yes	1.05 (0.85–1.28)		0.95 (0.77–1.17)	
Peer isolation	No	Ref		Ref	Ref
	Yes	1.60 (1.37–1.86)		1.31 (1.10–1.54)	1.32 (1.10–1.58)
Lower household income (<3,000,000)	No	Ref		Ref	Ref
	Yes	1.49 (1.25–1.77)		1.51 (1.24–1.83)	1.15 (0.90–1.47)

Note. All analyses were weighted for grade. Model 1 adjusted child’s sex, grade, number of sibling, maternal age, and child’s schools. Model 2, which examined the association of each type of ACE with playing time of game independently, adjusted child’s sex, grade, number of sibling, maternal age, and child’s schools. Model 3 added type of ACE which were significant in Model 2. ACE = adverse childhood experience, OR = odds ratio, 95%CI = 95% confidence interval.

## Data Availability

The data presented in this study are available on request from the corresponding author. The data are not publicly available due to ethical restrictions.

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
