# Peer review of "Association between Adverse Childhood Experiences and Time Spent Playing Video Games in Adolescents: Results from A-CHILD Study"

_ijerph, 2021, doi:10.3390/ijerph181910377_

Round 1

Reviewer 1 Report

Please check for spelling errors throughout the manuscript (e.g. 44-46, 69-70, 122-123 etc.)

I was wondering, why all questionnaires with at least JUST one answer were deemed valid?! It seems to be a somwhat too liberal inclusion criteria.

It seems rather arbitrary to equal "3 hours or more" with 210 minutes (=3,5 hours) for all participants. The authors may prove the applicability of a rather more evidence driven approach, given the somehwat linear decrease of cases responding "yes" to each of the different categories presented. Thus, it may be more plausible to model a linear trend based on the case frequencies per 30 minute classes to approximate a more plausible average duration of gaming minutes per day for those people responding  > 3 hours.

In addition, the authors should provide a revised result for the overall average gaming time per day based on that, also excluding those who don't game at all, just based on those participants that game.

I recommend to split the left and the right half of table 2 into separte tables (2a & 2b). This would make it much more easier to read and understand. In doing so, an column with all wordings of categories should be added to table 2b.

The authors should provide the number of the ethical approval.

Please correct "Schmahl" instead of "Scmhl" ( l. 230)

Please check that all references are provided in accordance to the submission guidelines and add any missing information:

  • page numbers of some references are missing (e.g. R2)
  • journal titles of some references are abbreviated (e.g. R15)
  • paper and journal titles of some refernces have no capitalized letters (e.g.  R9)

Reviewer 2 Report

I deem the topic of the submitted manuscript relevant and of potential interest to the reader and scientific community. However there are two main issues which induce me not to find the paper convincing. The first concern is about the methods. The paper does not explain clearly the study design (who answers what - caregivers and children - and how then responses are linked to each single record) as well as the type of measurement instrument used. As far as I can understand from the methods section (2.2. Measurements), which by the why I found rather obscure, the authors state that they used the ACE scale. Premising that I did not know this scale, I looked for it, and specifically to the revised version, quoted as the main reference (Finkelhor et al., 2015). In that version, on the top of the original 10 items, four other items were proposed, constructed as a dummy variable with a score of 0 if the youth never experienced it and a code of 1 if he or she experienced it at any time in his or her life. The questionnaire was administered to youth (who are also the potential victims of the abuses/violence investigated). As far as I could understand, the authors did not use the ACE scale and the questions were directed to the caregiver instead of the children. I found this rather strange because I would expect that the potential victims should report any parental abuse, rather than parents themselves. I found reference to the Isumi et al., 2018 paper which seems to be based on the same instrument. But I think that the methods used should always be made very clear in the methods sections, which I do not believe was done in the present paper. In the Results section, Tables should always be self-explanatory and consistent, which I don't think it is the case (in Tab 1 N and % are reported in two columns, in Tab 2 % into parentheses; in Tab 2 notes are missing, and the table is obscure (column titles are just numbers of ACEs?); in Table 3 the ACE total score is (0-8)?). The second concern is about the results. First I did not see a clear dose-response relationship between the investigate factors. Second the findings do not seem to be very promising. Even in the case of 8.47 minutes more per day playing video games, is this so relevant? In my opinion it is not a lot, and if authors think this is a relevant finding they should explain why and support it with relevant literature. Even the conclusions do not provide any relevant policy indication.

Reviewer 3 Report

This study included an adequate sample. The measure is valid. The analysis is reasonable. The result was well presented and demonstrated important results. Its result also has a good implication in the prevention of gaming disorder. 

I might suggest that introduction and discussion could provide some information about gaming disorder. 
